# Piloting *Siyakhana*: A community health worker training to reduce substance use and depression stigma in South African HIV and TB care

Kristen S. Regenauer[1]*, Alexandra L. Rose[1], Jennifer M. Belus[2,3], Kim Johnson[4], Nonceba Ciya[4], Sibabalwe Ndamase[4], Yuche Jacobs[4,5], Lexy Staniland[6], Goodman Sibeko[7], Ingrid V. Bassett[8], John Joska[7], Bronwyn Myers[4,6,7‡], Jessica F. Magidson[1,9‡]

**1** Department of Psychology, University of Maryland, College Park, College Park, Maryland, United States of America, **2** University of Basel, Basel, Switzerland, **3** Department of Clinical Research, University Hospital Basel, Basel, Switzerland, **4** Mental Health, Alcohol, Substance Use and Tobacco Research Unit, South African Medical Research Council, Parow, Cape Town, South Africa, **5** People Development Centre: Corporate Wellness, Western Cape Department of Health & Wellness, Plumstead, Cape Town, South Africa, **6** EnAble Institute, Faculty of Health Sciences, Curtin University, Perth, Australia, **7** Department of Psychiatry and Mental Health, University of Cape Town, Cape Town, South Africa, **8** Division of Infectious Diseases, Medical Practice Evaluation Center, Massachusetts General Hospital/Harvard Medical School, Boston, Massachusetts, United States of America, **9** Center for Substance Use, Health & Addiction Research (CESAR), University of Maryland, College Park, College Park, Maryland, United States of America

‡ BM and JFM are joint senior authors to this work.
* kregenau@umd.edu

**Data Availability Statement:** De-identified data from this study has not been deposited into an open-access repository because it was not

## Abstract

South Africa has one of the highest rates of HIV/tuberculosis (TB) co-infection, and poor engagement in HIV/TB care contributes to morbidity and mortality. In South Africa, community health workers (CHWs) are tasked with re-engaging patients who have dropped out of HIV/TB care. CHWs have described substantial challenges with substance use (SU) and depression among their patients, while patients have described CHW stigma towards SU and depression as barriers to re-engagement in care. Yet, CHWs receive little-to-no training on SU or depression. Therefore, we piloted *Siyakhana*, a brief CHW training to reduce stigma related to SU and depression while improving skills for re-engaging these patients in HIV and/or TB care. This study evaluated the preliminary effectiveness (stigma towards SU and depression; clinical competence assessed via roleplay) and implementation (quantitative ratings of feasibility, acceptability, appropriateness, adoption; semi-structured written qualitative feedback) of *Siyakhana* among CHWs and supervisors ($N$ = 17) at pre- and post-training assessments. SU stigma significantly decreased ($F(1,16)$ = 18.94, $p$ < 0.001, $\eta_p^2$ = 0.54). Depression stigma was lower than SU stigma at both timepoints and did not significantly decrease after training. CHW clinical competency towards patients with SU/depression significantly improved ($t(11)$ = -3.35, $p$ = 0.007, d = 1.00). The training was rated as feasible, acceptable, appropriate, and likely to be adopted by CHWs and their supervisors. Nonjudgmental communication was commonly described as the most useful training

included in participants' consent forms. De-identified study materials, including the analysis code, are available from Dr. Michael Wagner (wagner@umd.edu) upon written request. Dr. Wagner is a consulting biostatistician in the Global Mental Health and Addiction Program (GMAP) at the University of Maryland, College Park and Director of Computer Operations at the Center for Substance Use, Addiction, & Health Research (CESAR) at the University of Maryland, College Park.

**Funding:** This study was funded by the National Institute of Mental Health (NIMH) grant R34MH122268 (awardees: JFM, BM). Additional support was received from the National Institute on Drug Abuse (NIDA) grant R21DA053212 (awardees: JFM, BM). KSR's time was supported by the National Institute on Drug Abuse (NIDA) grant R36DA057167, ALR's time was supported by NIMH grant F31MH123020, and JMB's time was supported by the Swiss National Science Foundation (grant no. PZ00P1_201690). The funders had no role in study design, data collection and analysis, decision to publish, or preparation of the manuscript.

**Competing interests:** The authors have declared that no competing interests exist.

component. Based on this pilot, the training is being refined and evaluated in a larger randomized stepped-wedge clinical trial.

## Introduction

Challenges with long-term engagement in care substantially contribute to HIV morbidity and mortality worldwide. Tuberculosis (TB) is a leading cause of death among people living with HIV, including in South Africa, the country with the greatest number of people living with HIV [1] and one of highest rates of HIV/TB co-infection [2]. As adherence to both HIV medication (i.e., antiretroviral therapy (ART)) and TB medication is required to prevent death in those co-infected [2], strategies to retain and re-engage people in care for both of these infectious diseases is critical. Although South Africa has the largest antiretroviral therapy (ART) program globally [1], and most individuals with TB are initiated on treatment [3], one-third of people living with HIV have not achieved viral suppression [1] and around 50% of people with TB do not complete treatment [3]. To help address this issue, community health workers (CHWs)—lay health workers who live in or around the community in which they serve—are relied on to visit patients' homes with the goal of facilitating retention or re-engagement in care.

Concurrently, it is estimated that South Africa has high rates of problem substance use (SU) [4–7] and depression [8], including among people living with HIV and TB [9–14]. For instance, a recent South African study examining patients with TB in primary care found that 83% screened positive for a psychiatric diagnosis, with depression among the most common diagnoses, and 43% screened positive for a SU disorder. Over half of these patients were concurrently living with HIV, with similar rates of psychiatric and SU diagnoses identified in these patients [13]. In other studies, rates of depression have been estimated to be as high as 41% among people with HIV [9] and 64% among people with TB in South Africa [14]. While there is limited data on the rates of SU among people with HIV and TB, previous work from our team found nearly one-third of people initiating ART had unhealthy alcohol use [11], and 88.4% of people initiating TB treatment had unhealthy alcohol use or smoked illicit drugs [12].

As SU and depression are associated with worse HIV and TB outcomes [15, 16], such as more missed visits and greater likelihood of being lost to follow-up [17, 18], it is likely that many of the patients who are visited by CHWs for recent disengagement in HIV/TB care experience symptoms of depression and/or SU. Yet, while the CHW role is meant to facilitate care engagement, research suggests that CHW stigma towards SU and depression can act as a barrier to engagement in HIV and/or TB care [19–24]. For example, in previous qualitative work, patients living with chronic conditions like HIV who were experiencing common mental health conditions (e.g., depression, substance use) shared not wanting to return to care because of how providers, including CHWs, treated them [20, 21], and worried that CHWs would not be appropriate people to screen for and intervene with common mental health conditions because of CHWs' negative attitudes towards such conditions [22]. Currently, CHWs receive little-to-no training on common mental health conditions like SU or depression, including how to assess for symptoms, refer for treatment, or effectively communicate with such patients [25]. In the Western Cape province, CHWs receive standard basic training on managing chronic care (e.g., for HIV, TB, diabetes), and their only education requirement before being appointed is the ability to read and write. While limited mental health training for CHWs has been piloted in this context, this has not yet been scaled up [26]. This lack of training on SU and depression limits CHWs' capacity to respond to these needs among their patients. Further,

lack of knowledge around depression and SU can contribute to stigmatizing beliefs, which may affect how CHWs interact with and treat patients [26].

Consequently, CHW trainings that target reducing stigma towards patients with SU or depression symptoms may contribute to improved patient engagement in HIV/TB care. Yet, there remains a lack of training programs in mental health geared towards reducing stigma, including SU stigma, for non-specialist providers. There is a need for effective trainings that can be successfully implemented, especially in low-resource settings where non-specialist providers are increasingly relied on to help reduce the mental health and SU treatment gap [26–28].

This pilot feasibility study aimed to evaluate the preliminary effectiveness and implementation of a pilot training program designed to reduce stigma associated with SU and depression among CHWs working in HIV/TB care in the Western Cape, South Africa, with the goal of further refining the training for a larger stepped-wedge randomized trial. Formative qualitative interviews with patients, CHWs, other healthcare workers, and policymakers guided the development of this training. In the present study, we examined the training's preliminary effectiveness in reducing stigma related to SU and depression and in increasing clinical competency, and early implementation outcomes (i.e., acceptability, appropriateness, feasibility, and intent to adopt using both quantitative and qualitative feedback) [29].

## Methods

### Ethics statement

This study was approved by the Human Research Ethics Committee at the South African Medical Research Council (SAMRC; protocol #EC039-10/2021) and by the City of Cape Town. All participants provided informed written consent prior to study participation. The study was conducted in accordance with the Declaration of Helsinki, the South African Guidelines for Good Clinical Practice, and South Africa's Protection of Personal Information (POPI) Act.

### Setting and participants

In the Western Cape, non-governmental organizations (NGOs) are contracted by the City of Cape Town and Western Cape Department of Health to employ CHWs to provide basic health support to publicly funded primary care clinics. CHWs are generally linked to a specific health clinic and visit the homes of patients with various health concerns and needs. Common CHW tasks include delivering medications to patients and providing daily observed therapy for TB, providing patients with basic health education, and following-up with patients with HIV and/ or TB who have disengaged from HIV or TB care. In the present study, our team partnered with TB HIV Care and Kheth'Impilo, two NGOs that employ many of the CHWs in Khayelitsha and Eastern health subdistricts in Cape Town, where the study was based. Between November 2021 and February 2022, CHWs ($n = 10$) and their supervisors ($n = 7$), all of whom were nurses, were recruited from the Eastern and Khayelitsha health subdistricts in the Cape Town metropole of the Western Cape province of South Africa, both of which serve predominately low-income populations [30, 31]. The purpose of this pilot study was to refine the training for a larger stepped wedge randomized clinical trial.

Following approval by the City of Cape Town, the study team approached site managers who arranged meetings between the study team and potential participants. Eligibility criteria included: (1) currently employed as a CHW or CHW supervisor at a partner NGO; (2) work role included supporting patient re-engagement in HIV or TB care; (3) available for a 2.5-day training; and (4) able and willing to complete informed consent and study procedures in English, isiXhosa, or Afrikaans (i.e., the three primary languages spoken in the province).

## Study procedures

Prior to the training, all participants provided written informed consent to participate in the training. All participants were assigned a study identification (ID) number based on the order in which they were enrolled; only staff working directly with participants at the South African Medical Research Council had access to the link between study ID numbers and participant names. Following informed consent procedures, participants completed a pre-training assessment with a study staff member consisting of self-report demographic, job, and stigma assessments. Participants also completed a brief, videorecorded roleplay where they were instructed to "meet" with a new patient (played by a staff member) who was experiencing symptoms of problem SU and depression. This staff member was minimally involved in the training, was from the community and familiar both with the patient population and mental health intervention training, and was trained in the roleplay methodology by a US-based team member (ALR) based on research best practices [32]. Please see S1 File for an overview of the roleplay instructions. Role plays were conducted in English and followed standardized procedures established in prior work with lay health workers (described below) [32, 33]. Participants attended the CHW training after completing the pre-training assessment. The training was conducted in English (in which all participants were proficient), with training staff clarifying concepts in isiXhosa or Afrikaans as needed. Immediately following the training, all participants completed a post-training assessment of stigma, clinical competency, and implementation outcomes (described below). Self-report de-identified data was collected on REDCap electronic data captured tools hosted at the University of Maryland, College Park [34, 35]. Roleplays were saved on Box through the University of Maryland, College Park, and labeled with participants' study ID numbers. Altogether, three separate 2.5-day trainings were conducted: two with CHWs ($n$ = 5 per group), and one with supervisors ($n$ = 7). All trainings were delivered by two bilingual psychological counsellors registered with the Health Professions Council of South Africa, and one bilingual peer counsellor, all of whom were employed as study staff.

Pre-training assessments were conducted at either the clinic where a participant worked or an SAMRC satellite research office; all trainings and post-training assessments were conducted at the SAMRC satellite research office. The study flow is illustrated in Fig 1.

**Siyakhana training.** *Siyakhana*, which translates to "we build each other up" in isiXhosa, is a training for CHWs and their supervisors designed to reduce stigma surrounding SU and depression and teach skills for engaging patients with SU and depression symptoms in TB/HIV care. *Siyakhana's* design was informed by Link and Phelan's stigma framework [36] and the Situated Information Motivation Behavioral Skills Model of Care Initiation and Maintenance framework (sIMB-CIM) [37]. According to the Link and Phelan stigma framework, mental health and SU stigma may exist among CHWs, when a) CHWs label their patients who have SU and/or depression symptoms as different and attach negative stereotypes to them, and b) separate these patients ('them') from other patients and themselves ('us'), leading these patients to experience status loss, discrimination, and conditional access to care. According to the sIMB-CIM model, engagement and maintenance in care is determined by a) accurate information about one's illness; b) intrapersonal and interpersonal motivation; and c) behavioral skills, including systems navigation and organizational/planning skills, all of which may be affected by SU and depression symptoms. We integrated these two models to conceptualize our understanding of training components that may reduce stigma and provide skills for engaging individuals with SU and depression in TB/HIV care. In this integrated model, CHWs would be provided with information to increase understanding of depression and SU and reduce stigma (i.e., accurate psychoeducation on SU and depression; information on

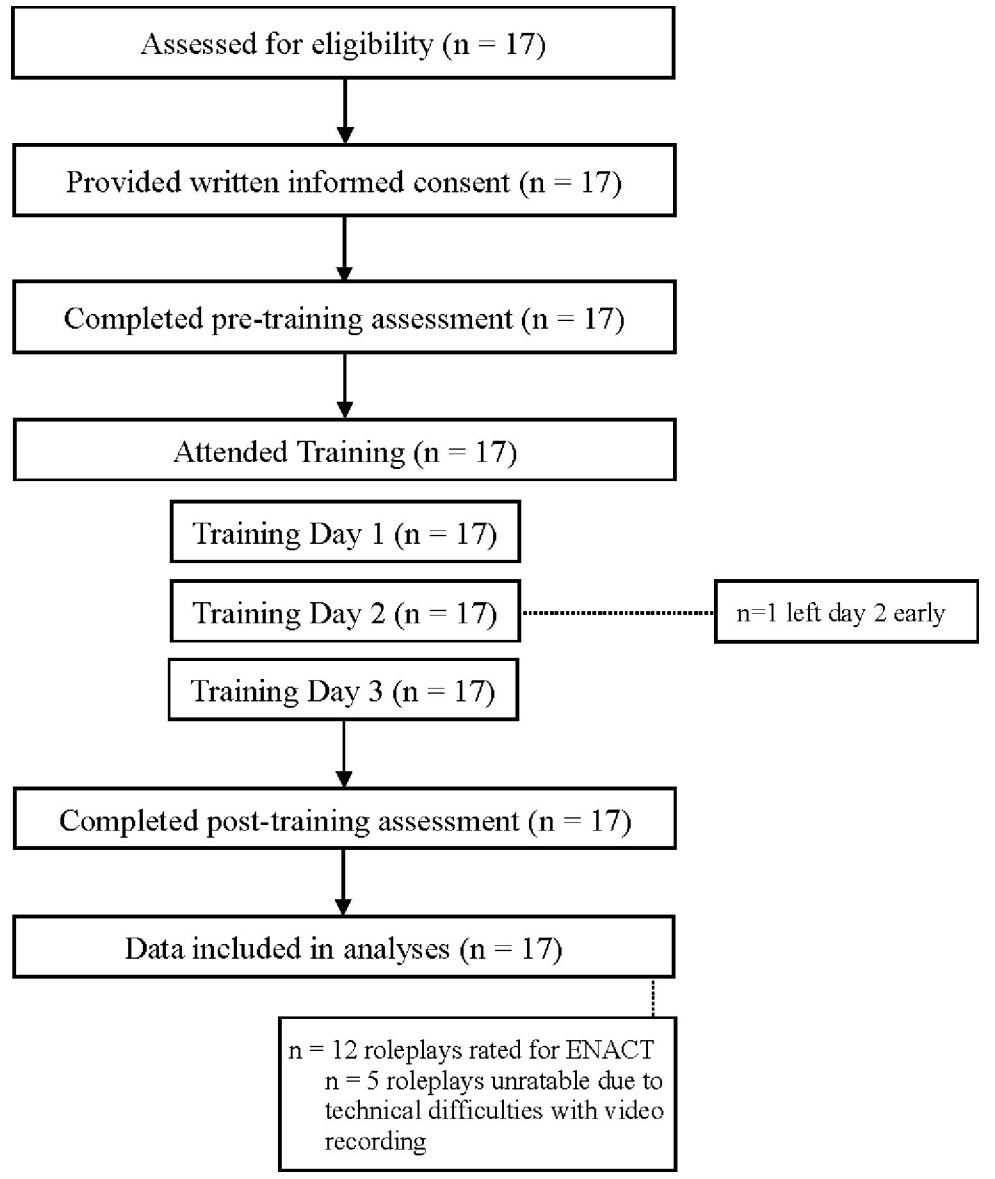

**Fig 1. Study flow.**

referrals and resources in community; information on stigma and why it can be harmful); taught non-judgmental communication skills to decrease stigmatized interactions with patients; asked to reflect on similarities between themselves and patients to decrease stigma towards patients; and taught motivational and problem-solving skills to help patients to better navigate and engage in care.

Specific content of the training was informed by formative qualitative interviews with patients receiving HIV/TB care, CHWs, healthcare workers, NGO leaders, and policymakers [38] and previous CHW training interventions to improve mental health literacy and skills for supporting patients who have symptoms of problem SU or depression [26, 39–43]. Please see S1 Table for a full list of training content. Suggestions from the qualitative interviews included framing the training as a way to empower CHWs in their roles; addressing cultural beliefs which may be held by some CHWs around addiction or mental health; focusing on core

motivational interviewing (MI) skills (i.e., open questioning, affirming, reflecting, and summarizing); including supervisors in the training; and including people with lived experience of SU or other mental health concerns in the training [38].

Based on the integrated conceptual model and formative interviews, *Siyakhana* includes four main components: (1) psychoeducation on depression, SU, stigma, HIV, TB, and how they intersect; (2) self-care strategies for CHWs (e.g., reflecting on one's background, identifying values, mindfulness); (3) evidence-based strategies for working with patients who may have depression or use substances (i.e., non-judgmental communication skills, components of MI, problem solving strategies); and (4) videos of real patients talking about their lived experience with mental health and SU challenges, along with reflection on who CHWs may know with similar challenges in their personal lives, as strategies for reducing social distance. The training was presented to participants with a toolbox analogy: the research team was equipping them with evidence-based tools that might help them work more efficiently and more effectively. The training utilized presentation, discussion, and roleplay methods to help participants learn each of the components.

**Inclusivity in global research.**   Additional information regarding the ethical, cultural, and scientific considerations specific to inclusivity in global research is included in supplemental materials (S2 File).

## Measures

Measures were collected at two timepoints: pre-training (i.e., immediately before first day of training) and post-training (i.e., immediately after the training had ended). At pre-training, participants were asked their age, gender, education level, primary language, job title, and length of time in their current position and occupation. They were also asked if they were from any of the locations where they currently worked and which aspects of their identity were the most important to them.

**Preliminary effectiveness (pre- to post-training).**   *Stigma*. Self-report stigma towards SU and depression was measured using a modified Social Distance Scale (SDS) [44], a common measure of mental health stigma. In this modified SDS, participants were given two case vignettes (i.e., short narratives about hypothetical characters in specific situations) about a patient with HIV: one presenting with symptoms of major depressive disorder, and one presenting with symptoms of SU disorder. Neither vignette directly stated that the patient had a diagnosis, and all symptoms were based on DSM-5 criteria [45]. The same vignettes were presented at both timepoints, and versions of both vignettes have previously been used in this setting [46, 47]. Participants rated their willingness to engage with each hypothetical patient across six social interactions, ranging from having a conversation with someone like the patient to marrying someone like the patient, on a 4-point scale from (1) "definitely" to (4) "definitely not". For each vignette, scores were summed to create an overall score that ranged from 6 to 24, with higher scores indicating a desire for greater social distance from the patient, and therefore, greater stigma.

*Clinical competency*. Clinical skills were observed during standardized role plays and assessed with the ENhancing Assessment of Common Therapeutic Factors (ENACT) tool [32]. The ENACT is a widely used tool for assessing lay health worker competence in delivering mental health interventions and has previously been used in South Africa [33]. Fifteen competencies (e.g., non-verbal communication, empathy, explanation of confidentiality, collaborative goal setting) are rated across four levels (1 –harmful, 2 –some basic skills, 3 –all basic skills, 4 –advanced). Competencies that were missing from a roleplay, although the absence was not necessarily harmful, were coded as a "2". Competencies are listed in S1 Table.

As is standard with use of the ENACT, operationalization of each skill level was reviewed and minor adaptations were made to reflect the setting and training goals before rating (e.g., asking client about treatment goals should not focus only on HIV, but also attend to mental health and SU in the current study). Total ENACT scores range from 15 to 60, with higher scores indicating greater clinical competency. There is no standard overall cut-off for "competent," although a rating of 3 or 4 is considered "competent" on each individual item.

**Implementation.** *Implementation ratings.* Implementation ratings were measured at post-training using a validated quantitative measure, based on the RE-AIM implementation framework [48], and designed for evaluating lay health worker interventions and trainings in low- and middle-income countries [49]. The measure was adapted for this training and has previously been used in similar settings [50]. The measure has four subscales: acceptability (i.e., satisfying and agreeable), feasibility (i.e., one is able to participate given their current resources), appropriateness (i.e., fits one's needs), and adoption (i.e., one will be willing to try and continue). All items were rated on a four-point scale (0 = "not at all", 3 = "a lot"), and items in each subscale were averaged for a final subscale score. Using the same four-point scale, participants were also asked if they thought the training fit the definitions of adoption, acceptable, appropriate, feasible, and accessible (i.e., can be easily available to those who need it). Three questions related to supervision and transportation were removed from the analysis as supervision was not included in this initial pilot training, and transportation was provided.

*Training semi-structured qualitative feedback.* To explore participants' thoughts on the training, at the post-training assessment, participants were asked to provide written qualitative feedback based on three semi-structured probes: (1) what they thought was the most useful part of the training; (2) what they thought was the least useful part of the training; and (3) any other open-ended feedback they had on the training. Training components mentioned in questions 1 and 2 were compiled into a list, and the number of participants who stated each as the most-useful and least-useful components were counted. Suggestions for future iterations of the training mentioned in question 3 were also listed, with the number of participants who endorsed each counted.

## Data analyses

All quantitative analyses were performed in R (v4.2.2) [51]. As the purpose of this study was to refine the training for a larger pilot trial, the study was not powered to assess effectiveness, and all effectiveness findings should be considered preliminary. The packages stats and rstatix were used for analyses [51, 52], and ggplot2 and ggpubr were used to create figures [53, 54]. Descriptive statistics were calculated for all stigma and clinician competency scores at pre- and post-training. Descriptive statistics were also calculated for each implementation subscale and definition at post-training. For self-report stigma, a 2 (pre-training vs. post-training assessment) x 2 (SU vs. depression vignette) repeated measures ANOVA, blocked by participant, was conducted to compare stigma associated with different vignettes at different timepoints. If the time*vignette interaction was significant, a one-way repeated measures ANOVA was run, conditioned on each factor, to examine simple main effects. For clinical competency, a paired t-test was conducted to compare the overall competency score at pre- and post-test. All ANOVA and t-test assumptions were checked prior to conducting analyses, including examining a QQ plot of the ANOVA models' residuals and conducting a Shapiro-Wilk test of normality.

For training feedback, two coders (KSR, LS) independently reviewed all open-ended feedback responses and recorded: (1) most useful training components; (2) least useful training components; (3a) whether a response contained a suggestion or desire for future trainings;

and (3b) if so, the suggestions. Results were compared and there were no discrepancies. De-identified data, including the analysis code, are available upon written request to the study principal investigators.

## Results

Table 1 shows the demographic and job characteristics of all participants. Participants were mostly female (94%), Black African (76%) or Coloured (South African racial category; 24%) and were an average of 47.5 years old ($SD$ = 9.8). Religion was the most frequently endorsed important aspect of participants' background/ identity (endorsed by 88%), followed by gender (endorsed by 82%), community of origin (endorsed by 65%), and ethnic group (endorsed by 53%). Ninety percent of CHWs and 29% of supervisors reported currently working in their community of origin. Regarding previous experience with SU or depression, 57% of CHW supervisors but only 10% ($n$ = 1) of CHWs reported that they had ever received previous training on depression or SU, and 90% of CHWs but only 43% of supervisors endorsed having someone with depression or problem SU in their family.

### Preliminary effectiveness (pre- to post-training)

**Stigma.** Descriptive statistics for stigma scores at both timepoints are presented in Table 2. The two-way repeated measure ANOVA revealed a significant interaction between vignette and timepoint, $F(1,48)$ = 6.79, $p$ = 0.008. Stigma towards SU significantly decreased between pre-training and post-training ($F(1,16)$ = 18.94, $p<0.001$, $\eta_p^2$ = 0.54), but stigma towards depression did not ($F(1,16)$ = 0.66, $p$ = 0.43, $\eta_p^2$ = 0.04). Stigma was consistently higher towards SU than depressive symptoms at both pre-training ($F(1,16)$ = 37.09, $p<0.001$, $\eta_p^2$ = 0.70) and post-training ($F(1,16)$ = 8.73, $p$ = 0.009, $\eta_p^2$ = 0.35) timepoints. Results are illustrated in Fig 2. The ANOVA table is presented in S2 Table.

**Clinical competency.** Descriptive statistics are presented in Table 2. Due to technical issues with recording equipment, complete pre-post role play pairings were only available for 12 of the 17 participants. A paired $t$-test of ENACT scores revealed that clinical competency scores significantly increased between pre- and post-training ($t(11)$ = -3.35, $p$ = 0.007, d = 1.00) (see Fig 3 and S3 Table). The median and interquartile (IQR) score ranges were 28.0 (24.0, 30.5) at pre-training, and 31.0 (29.8, 31.8) at post-training (possible range 15–60).

### Implementation

**Implementation ratings.** Acceptability, appropriateness, and adoption subscale scores were available for all $N$ = 17 participants; due to missing responses feasibility scores were available for $n$ = 16 participants only. The quantitative assessment indicated that participants rated the *Siyakhana* training to be highly acceptable ($M$ = 2.97, $SD$ = 0.07), feasible ($M$ = 2.71, $SD$ = 0.24), appropriate ($M$ = 2.89, $SD$ = 0.17), and suitable for adoption ($M$ = 2.94, $SD$ = 0.24). Participants also strongly agreed that the training aligned well with the definitions of adoption ($M$ = 2.94, $SD$ = 0.24), acceptable ($M$ = 2.94, $SD$ = 0.24), appropriate ($M$ = 2.82, $SD$ = 0.39), feasible ($M$ = 2.88, $SD$ = 0.33), and accessible ($M$ = 2.88, $SD$ = 0.49). These findings are presented in Table 2.

**Training feedback.** Training feedback is summarized in Table 3. The following training components were reported as most useful: non-judgmental communication ($n$ = 9 endorsed), information on depression ($n$ = 7), problem solving ($n$ = 6), mindfulness ($n$ = 6), information on SU ($n$ = 5), motivational interviewing ($n$ = 5), confidentiality ($n$ = 5), information on culture ($n$ = 3), information on stigma ($n$ = 3), information on TB and HIV ($n$ = 1), and general self-care skills ($n$ = 1). Most participants ($n$ = 13) did not report any components as "least

**Table 1. Demographic & job characteristics.**

| Characteristic | Full Sample N = 17 | | Disaggregated By Role | | | |
|---|---|---|---|---|---|---|
| | | | CHW n = 10 | | Supervisor (Nurse) n = 7 | |
| | n (%) | | n (%) | | n (%) | |
| Gender | | | | | | |
| Cis-Woman | 16 (94%) | | 10 (100%) | | 6 (86%) | |
| Cis-Male | 1 (6%) | | - | | 1 (14%) | |
| Race[a] | | | | | | |
| Black African | 13 (76%) | | 9 (90%) | | 4 (57%) | |
| Coloured | 4 (24%) | | 1 (10%) | | 3 (43%) | |
| Primary Language | | | | | | |
| Xhosa | 12 (70%) | | 9 (90%) | | 3 (43%) | |
| Afrikaans | 4 (24%) | | 1 (10%) | | 3 (43%) | |
| Zulu | 1 (6%) | | | | 1 (14%) | |
| Highest Education | | | | | | |
| Did not complete high school | 6 (35%) | | 6 (60%) | | - | |
| Completed high school | 5 (29%) | | 4 (40%) | | 1 (14%) | |
| Any education post high school | 6 (35%) | | - | | 6 (86%) | |
| Most important aspects of background/ identity [a] | | | | | | |
| Religion | 15 (88%) | | 9 (90%) | | 6 (86%) | |
| Gender | 14 (82%) | | 9 (90%) | | 5 (71%) | |
| Community of Origin | 11 (65%) | | 6 (60%) | | 5 (71%) | |
| Ethnic Group | 9 (53%) | | 6 (60%) | | 3 (43%) | |
| Race | 8 (47%) | | 4 (40%) | | 4 (57%) | |
| Sexuality | 7 (41%) | | 5 (50%) | | 2 (29%) | |
| Years in Current Role | | | | | | |
| <1 year | 2 (12%) | | - | | 2 (29%) | |
| 1–5 years | 7 (41%) | | 4 (40%) | | 3 (43%) | |
| 5+ years | 8 (47%) | | 6 (60%) | | 2 (29%) | |
| Working in Community of Origin | 11 (65%) | | 9 (90%) | | 2 (29%) | |
| Any previous mental health (or SU) training | 5 (29%) | | 1 (10%) | | 4 (57%) | |
| Someone with depression or SU disorder in family | 12 (70%) | | 9 (90%) | | 3 (43%) | |
| | M (SD) | Range | M (SD) | Range | M (SD) | Range |
| Age | 47.5 (9.8) | 30.0–69.0 | 43.0 (6.8) | 30.0–50.0 | 54.0 (10.2) | 40.0–69.0 |
| Weekly Caseload | 24.6 (16.1) | 3.0–57.0 | 31.9 (16.2) | 3.0–57.0 | 14.1 (9.0) | 3.0–30.0 |

[a]Could select more than one option

useful." The remaining participants identified information on TB (*n* = 2), information on culture (*n* = 1), information on HIV (*n* = 1), and further clarifying the CHW role (*n* = 1) as the least useful components.

Seven participants included at least one suggestion for improving the training. These suggestions included offering more training (i.e., offering refresher courses or making the training

**Table 2. Outcome variable descriptive statistics.**

| Outcome | N | M (SD) | Median (IQR) | Range |
|---|---|---|---|---|
| Preliminary Effectiveness | | | | |
| Stigma | | | | (6–24 Possible) |
| Depression | | | | |
| Pre-Training | 17 | 8.12 (2.03) | 7.00 (7.00, 9.00) | 6–13 |
| Post-Training | 17 | 8.71 (2.44) | 8.00 (7.00, 10.00) | 6–14 |
| SU | | | | |
| Pre-Training | 17 | 14.00 (3.81) | 14.00 (12.00, 17.00) | 6–20 |
| Post-Training | 17 | 11.18 (3.43) | 10.00 (9.00, 13.00) | 6–18 |
| Clinical Competency | | | | (15–60 possible) |
| Pre-Training | 12 | 27.33 (4.14) | 28.00 (24.00, 30.50) | 20–33 |
| Post-Training | 12 | 31.42 (3.94) | 31.00 (29.75, 31.75) | 25–39 |
| Implementation | | | | |
| Scales | | | | (0–3 Possible)[b] |
| Adoption | 17 | 2.94 (0.14) | 3.00 (3.00, 3.00) | 2.50–3.00 |
| Acceptability | 17 | 2.97 (0.07) | 3.00 (3.00, 3.00) | 2.75–3.00 |
| Appropriateness | 16[a] | 2.88 (0.16) | 2.92 (2.83, 3.00) | 2.38–3.00 |
| Feasibility | 16[a] | 2.71 (0.24) | 2.75 (2.67, 2.85) | 2.08–3.00 |
| Definitions | | | | (0–3 Possible)[b] |
| Adoption | 17 | 2.94 (0.24) | 3.00 (3.00, 3.00) | 2.00–3.00 |
| Acceptable | 17 | 2.94 (0.24) | 3.00 (3.00, 3.00) | 2.00–3.00 |
| Appropriate | 17 | 2.82 (0.39) | 3.00 (3.00, 3.00) | 2.00–3.00 |
| Feasible | 17 | 2.88 (0.33) | 3.00 (3.00, 3.00) | 2.00–3.00 |
| Accessible | 17 | 2.88 (0.49) | 3.00 (3.00, 3.00) | 1.00–3.00 |

[a]One participant did not respond or responded "don't know" to an item on this measure

[b]Implementation Scoring: 0 –"Not at all" feasible, acceptable, etc. | 3 –"A lot" feasible, acceptable, etc.

longer; $n = 4$), training more people ($n = 2$), including more content on depression and substance use ($n = 1$), and conducting roleplays in the local language instead of in English ($n = 1$).

## Discussion

Findings suggest that immediately after the 2.5-day *Siyakhana* training for CHWs and their supervisors in the Western Cape, South Africa, stigma towards SU (but not depression) decreased and clinical competency for engaging with patients with symptoms of SU and depression improved. Participants rated the training as feasible, acceptable, appropriate, and likely to be adopted. Training feedback suggested that participants found the training useful. Notably, self-report stigma towards depression was significantly lower than SU stigma scores at both timepoints. This finding is consistent with other literature suggesting that SU is among the most stigmatized health conditions [46, 55–61]. For this reason, the training content and role plays focused more on SU than on depression, which may explain why a significant reduction was observed for SU stigma and not depression stigma. Further, the null finding could be due to a floor effect, as depression stigma scores were already very low at the pre-training assessment. For instance, nearly 18% of participants ($n = 3$) received the lowest stigma score possible at this first assessment.

   Although stigma towards SU did significantly decrease after training, the fact that depression stigma scores remained significantly lower than SU stigma scores suggest that CHWs

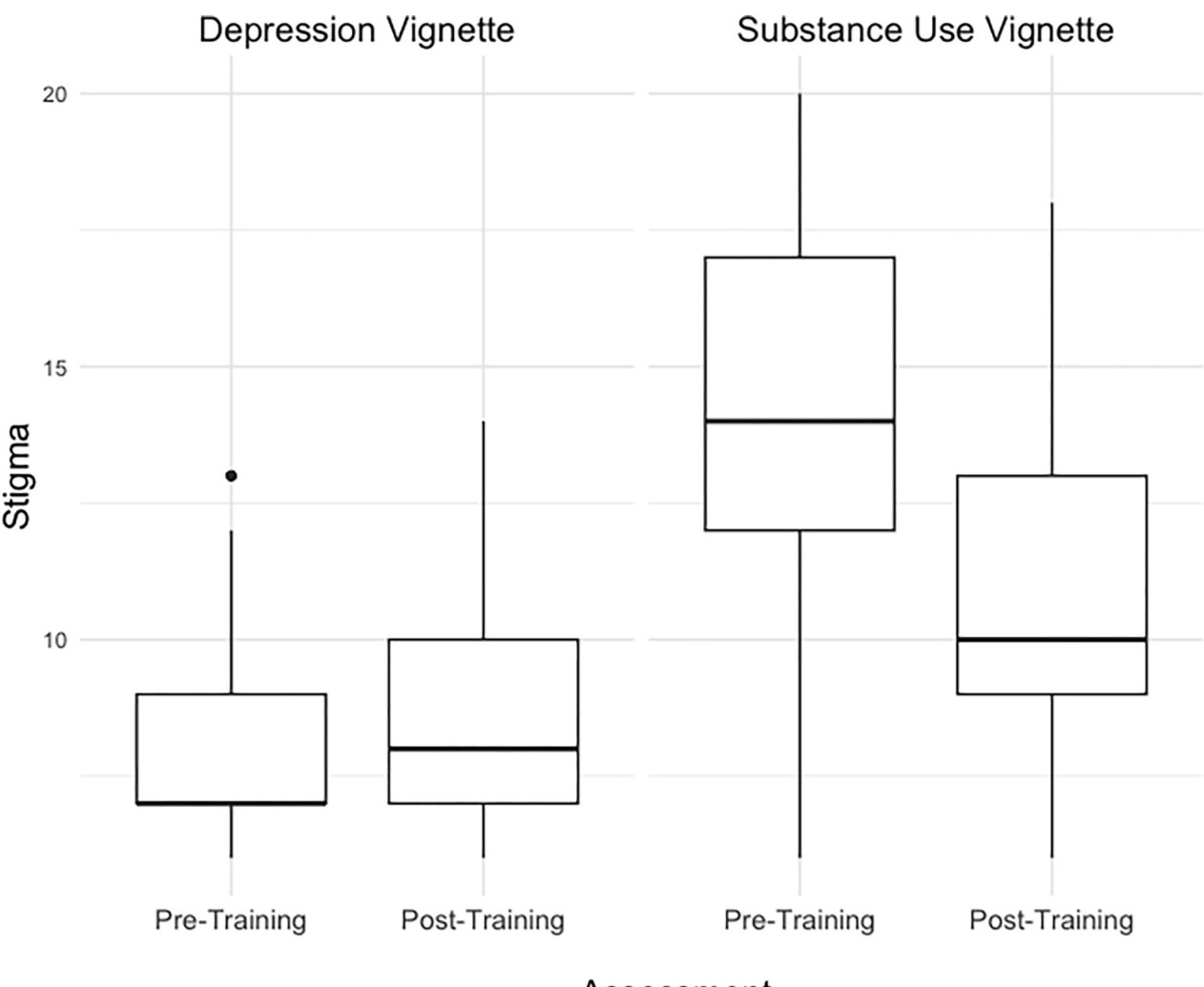

**Fig 2. Stigma scores as measured with the SDS across assessments and vignettes.**

could benefit from further SU training and support. Future trainings may want to consider increased contact with people with lived experience of SU disorder, as such contact has been shown to reduce stigma [62–64]. The present training was only able to expose participants to pre-recorded videos of patients with lived experience of SU disorder and depression symptoms due to in-person capacity constraints associated with the COVID-19 pandemic. Although most people endorsed having a family member with depression or SU, baseline SU stigma levels were quite high. One explanation for this was we asked people whether they had anyone with depression or problem SU in their family, rather than differentiating the two. Thus, it is possible most people had someone with depressive symptoms, not symptoms of SU disorder, in their family. Second, contact in the context of a training aimed at increasing understanding of these conditions and teaching skills for working effectively and empathetically with such patients may be more beneficial for reducing stigma than just any type of contact. Finally, ensuring this brief training is followed by supervision to support case management and enhance clinical competency, and training existing supervisors in such supervision so it continues to be ongoing after the study, may lead to further reductions in stigma and increase utilization of nonjudgmental communication and other skills learned in the training [65, 66].

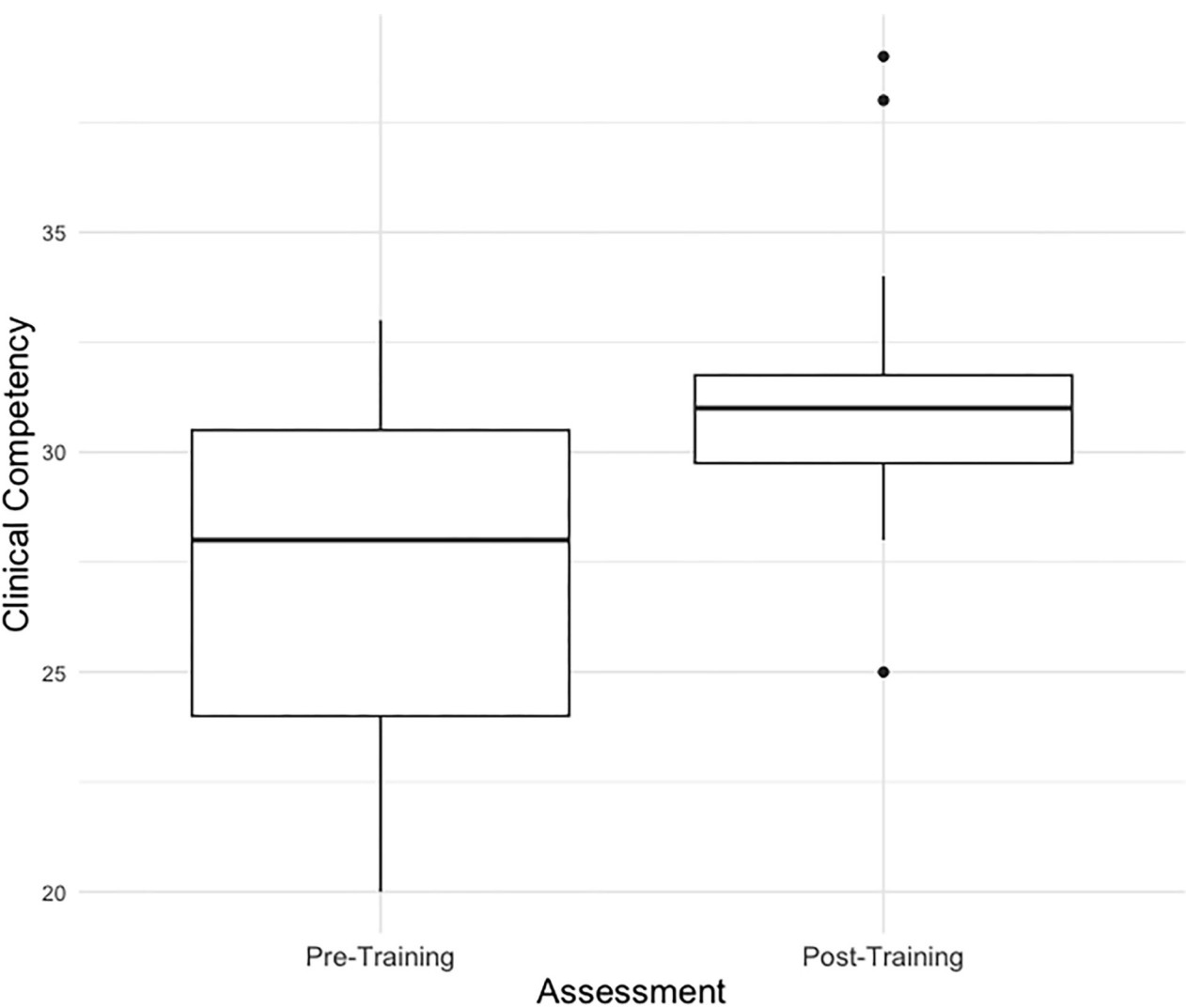

**Fig 3. Clinical competency scores as measured with the ENACT across assessments.**

As hypothesized, total ENACT scores improved post-training, suggesting that the training improved overall clinical competency for engaging with patients with symptoms of SU disorder/ depression. Conducting the ENACT in isiXhosa, as suggested by one participant in open-ended feedback, would be important in follow-up work.

Participants rated *Siyakhana* highly in feasibility, acceptability, appropriateness, and intentions to adopt the training. All participants completed the full training, suggesting that attending the training over three workdays was acceptable for their NGO and feasible for them. However, it should be noted that one participant felt emotionally distressed when discussing mental health on the second day of training and decided to leave early. This participant returned for the third day of training, debriefed with the registered counsellor facilitator, and was offered referral to mental health care services. Like their patients, CHWs live in communities with high exposure to trauma and other stressors, and low rates of mental healthcare access [67–69]. In fact, nearly all CHW participants endorsed having someone in their family with depression or a SU disorder. Therefore, it is important to consider that mental health trainings may bring up distress for participants, and to ensure suitable debriefing and support is available.

**Table 3. Training feedback.**

| Component | Most Useful | Least Useful |
|---|---|---|
| | *n* (%) | *n* (%) |
| Psychoeducation (Information) | 12 (71%) | 4 (24%) |
| Depression | 7 (41%) | - |
| Substance Use | 5 (29%) | - |
| Confidentiality | 5 (24%) | - |
| Culture | 3 (18%) | 1 (6%) |
| Stigma | 3 (18%) | - |
| TB | 1 (6%) | 2 (12%) |
| HIV | 1 (6%) | 1 (6%) |
| CHW Role | - | 1 (6%) |
| Evidence-Based Patient Skills | 13 (76%) | - |
| Non-judgmental Communication | 9 (53%) | - |
| *Non-Verbal Specified* | *3 (18%)* | - |
| *Verbal Specified* | *2 (12%)* | - |
| *Empathy/ Sympathy Specified* | *1 (6%)* | - |
| Problem Solving | 6 (35%) | - |
| Motivational Interviewing | 5 (29%) | - |
| *Affirmations Specified* | *2 (12%)* | - |
| *Reflective Listening Specified* | *2 (12%)* | - |
| *Open-Ended Questions Specified* | *2 (12%)* | - |
| Self-Care Skills | 7 (41%) | - |
| Mindfulness | 6 (35%) | - |
| Self-Care (Generally) | 1 (6%) | - |
| None | - | 13 (76%) |
| Suggestions | | *n* (%) |
| More training desired | | 4 (24%) |
| *5-day training specified* | | *2 (12%)* |
| *Refresher course specified* | | *1 (6%)* |
| Train more people | | 2 (12%) |
| *Train more CHWs specified* | | *1 (6%)* |
| *Train people in the health system and government specified* | | *1 (6%)* |
| Include more content on depression and substance use | | 1 (6%) |
| Conduct roleplays in local language | | 1 (6%) |
| No suggestions provided | | 10 (59%) |

NOTE: As answers were open-ended, participants could write more than one component.

While not formally measured, in conducting the training our team learned that CHWs are not taught about patient confidentiality, how to screen for SU or depression symptoms, or the possible referral pathways for SU or depression in patients. Therefore, the next iteration of this training will devote more time addressing confidentiality; include additional training material for screening for SU disorder, depression, and risk in patients; and include training on mental health and SU referrals in the community.

Training components reported as most useful included nonjudgmental communication, psychoeducation around depression specifically, problem-solving, and mindfulness. Notably, while other South African task-shared mental health trainings have included components like communication and problem-solving [39], mindfulness training has been relatively limited

[70, 71]. As mindfulness was also ranked as one of the most useful skills among people living with HIV who use substances in a previous intervention in a similar community [72], future trainings may want to include a mindfulness component.

There are several limitations to consider when interpreting findings. Most notably, this was a pilot study to inform a future Type 1 hybrid effectiveness-implementation study and had a small sample size. The present study included both CHWs ($n = 10$) and their supervisors ($n = 7$), all of whom were nurses and therefore had more healthcare training. As such, it is possible that results from this study would have been different had all participants been CHWs. However, based on formative qualitative interviews with patients, CHWs, healthcare workers, and policymakers, we believed it was essential to include CHW supervisors in this phase of the training to get their feedback on the training's format and content. Another limitation is that outcomes were only assessed immediately after the training. Consequently, the lasting effects of this training are undetermined, including the practical utility of the skills developed could not be assessed. Longitudinal research to examine maintenance and follow-up exploration of how participants use the skills in actual practice will be important for future evaluation of this training. Finally, participants completed all assessments in English with the help of study staff who delivered the training. Thus, it is possible that findings were impacted by language and/or social desirability bias. To minimize this risk, study staff who were bilingual in English and other commonly spoken languages (isiXhosa, Afrikaans) were available to help participants when needed. Participants were also reminded that answering honestly would help us better refine the training, that their answers were confidential, and that they would not be penalized or judged for any of their answers.

Despite the limitations, the current study had several strengths. First, rather than focusing on patients with just one stigmatized identity, participants were exposed to roleplays and vignettes of patients with multiple and intersecting stigmatized identities. As identities do not exist in a vacuum (i.e., no one is just a woman or just Black) [73] and different intersecting identities, conditions, and stigmas may affect the way CHWs view patients [21, 74–76], exposing CHWs to scenarios with complex, multifaceted patients may increase the effectiveness and generalizability of the training. Regarding implementation, the training was relatively brief. Compared to other CHW trainings to increase clinical competency for depression and SU that tend to be longer [26, 32, 77], the length of this training may make it more feasible to deliver at scale. Further, the training was successfully delivered by a small team consisting of two psychological counsellors, and a peer counsellor with personal lived experience. This suggests that with proper training and supervision, less specialized providers may be able to deliver this training, also increasing its feasibility and scalability.

Ultimately, this pilot study provides some of the first evidence that providing CHWs with relatively brief SU and depression training can significantly reduce SU stigma among this workforce and improve their competencies to interact with patients with SU and depression in the context of HIV and TB care. This is important as CHWs are increasingly being relied on to provide task-shared psychological interventions in LMICs [78], including South Africa [79] and CHW stigma towards mental health and SU can affect patient engagement in care [19–24]. Findings from this pilot are being used to further refine the training and inform the next phase of the study, which is a larger, randomized stepped-wedge clinical trial with longer-term effectiveness and implementation outcomes (NCT05282173).

## Supporting information

**S1 File. ENACT roleplay staff training.**
(DOCX)

**S2 File. Inclusivity in global research.**
(DOCX)

**S1 Table. Training components.**
(DOCX)

**S2 Table. ENACT clinical competencies.**
(DOCX)

**S3 Table. ANOVA comparing SDS scores across vignettes and assessments.**
(DOCX)

**S4 Table. T-Test comparing total ENACT scores at different assessments.**
(DOCX)

## Acknowledgments

We acknowledge the rest of the Siyakhana–C team. We also acknowledge and thank the NGOs Kheth'Impilo and TB HIV Care for partnering with this study and being supportive of this training. We would especially like to thank study participants for their time, input, and contributions to this study.

## Author Contributions

**Conceptualization:** Bronwyn Myers, Jessica F. Magidson.

**Formal analysis:** Kristen S. Regenauer, Alexandra L. Rose, Nonceba Ciya, Sibabalwe Ndamase, Lexy Staniland.

**Funding acquisition:** Bronwyn Myers, Jessica F. Magidson.

**Investigation:** Kim Johnson, Nonceba Ciya, Sibabalwe Ndamase, Yuche Jacobs, Bronwyn Myers, Jessica F. Magidson.

**Methodology:** Bronwyn Myers, Jessica F. Magidson.

**Project administration:** Kristen S. Regenauer, Kim Johnson, Yuche Jacobs.

**Software:** Kristen S. Regenauer.

**Supervision:** Kristen S. Regenauer, Alexandra L. Rose, Jennifer M. Belus, Yuche Jacobs, Goodman Sibeko, Ingrid V. Bassett, John Joska, Bronwyn Myers, Jessica F. Magidson.

**Visualization:** Kristen S. Regenauer.

**Writing – original draft:** Kristen S. Regenauer, Alexandra L. Rose.

**Writing – review & editing:** Alexandra L. Rose, Jennifer M. Belus, Kim Johnson, Nonceba Ciya, Sibabalwe Ndamase, Yuche Jacobs, Lexy Staniland, Goodman Sibeko, Ingrid V. Bassett, John Joska, Bronwyn Myers, Jessica F. Magidson.

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
