## [Decision Letter · Decision Letter 0]

13 Aug 2023

PGPH-D-23-00892

Piloting Siyakhana: A community health worker training to reduce substance use and depression stigma in South African HIV and TB care

Dear Dr. Regenauer,

Thank you for submitting your manuscript to PLOS Global Public Health. After careful consideration, we feel that it has merit but does not fully meet PLOS Global Public Health’s publication criteria as it currently stands. Therefore, we invite you to submit a revised version of the manuscript that addresses the points raised during the review process.

EDITOR:

1. Kindly place the person before the disease and therefore replace terminologies such as 'TB patient' and 'HIV patient' with 'people with TB' and 'people living with HIV/AIDS' or similar wherever applicable.

2. While this study is a pilot as stated it the design Type 1 hybrid implementation study seems a bit premature for this study. You may consider (though not necessarily) an 'intervention study' or 'quasi-experimental feasibility pilot' as alternatives.

3. Kindly indicate if your study is powered adequately to assess effectiveness.

4. The study states short term effectiveness was assessed- however, the assessments for effectiveness were done only pre and immediately post intervention and not beyond, also the term preliminary effectiveness has been use elsewhere. Do consider the use of uniform terminology throughout the manuscript.

5. Please consider depositing your data in an open access repository- or alternatively indicating how this data may be made available to those interested.

6. I request you to kindly address the comments from the reviewers that have been stated below.

We look forward to receiving your revised manuscript.

Kind regards,

Rashmi Josephine Rodrigues, M.D., Ph.D.

Academic Editor

Journal Requirements:

2. Please include the following request in the decision letter, and ping me with follow-up. “Please include a complete copy of PLOS’ questionnaire on inclusivity in global research in your revised manuscript. Our policy for research in this area aims to improve transparency in the reporting of research performed outside of researchers’ own country or community. The policy applies to researchers who have travelled to a different country to conduct research, research with Indigenous populations or their lands, and research on cultural artefacts. The questionnaire can also be requested at the journal’s discretion for any other submissions, even if these conditions are not met.  Please find more information on the policy and a link to download a blank copy of the questionnaire here: https://journals.plos.org/globalpublichealth/s/best-practices-in-research-reporting. Please upload a completed version of your questionnaire as Supporting Information when you resubmit your manuscript.

3. Please provide separate figure files in .tif or .eps format only and remove any figures embedded in your manuscript file. Please also ensure all files are under our size limit of 10MB.

4. In the online submission form, you indicated that "De-identified study materials, including the analysis code, are available from the first author upon written request". All PLOS journals now require all data underlying the findings described in their manuscript to be freely available to other researchers, either 1. In a public repository, 2. Within the manuscript itself, or 3. Uploaded as supplementary information.

Additional Editor Comments (if provided):

Reviewers' comments:

Reviewer's Responses to Questions

**Comments to the Author**

1. Does this manuscript meet PLOS Global Public Health’s publication criteria? Is the manuscript technically sound, and do the data support the conclusions? The manuscript must describe methodologically and ethically rigorous research with conclusions that are appropriately drawn based on the data presented.

Reviewer #1: Yes

Reviewer #2: Yes

Reviewer #3: Yes

2. Has the statistical analysis been performed appropriately and rigorously?

Reviewer #1: Yes

Reviewer #2: Yes

Reviewer #3: Yes

3. Have the authors made all data underlying the findings in their manuscript fully available (please refer to the Data Availability Statement at the start of the manuscript PDF file)?

Reviewer #1: Yes

Reviewer #2: No

Reviewer #3: Yes

4. Is the manuscript presented in an intelligible fashion and written in standard English?

Reviewer #1: Yes

Reviewer #2: Yes

Reviewer #3: Yes

5. Review Comments to the Author

Reviewer #1: Dear Authors,

Stigma is an important area for evidence based interventions among people living with HIV. Your manuscript reads well and you have explained about the trainings and the tools used in the study well. Please find my comments below.

39-42: Here the patients with SU and depression are people living with HIV-TB co-infection? If yes, then authors may give some figures to show how much problem is SU/ depression in this group.

44: Instead of working, re-engaging would be a better word here since we are dealing with those patients who have dropped out of care.

60: Please add ‘long-term’ before the word engagement.

61-63: Please revise the sentence.

75: Authors may use mental health ‘conditions’ as used in the WHO mhGAP.

96-97: Authors may remove repetition of “preliminary effectiveness.”

104: It will be helpful for the readers if authors can provide a brief description of roles and responsibilities of CHWs and if they are based in the community or health facility.

Study procedures: Please mention who conducted these trainings? What was their expertise in conducting such trainings. It has been mentioned towards the conclusion in the manuscript. It will be good if some information is provided in the methods section as well.

162-63: “Specific content of the training was informed by formative qualitative interviews..” Can authors list the contents in a table?

194: What were the two time points? Before and after training? Please specify.

262: What is the meaning of ‘believed’ here? Not clear.

326-28: In table 1 authors have reported 90% of CHWs had a family member with SU/depression. They are already having contact with people with lived experience of SU. Inspite of this the baseline stigma scores for SU was high. How do the authors explain this?

342-49: This is very important to highlight. I am glad authors have discussed this in this manuscript.

Thank you!

Reviewer #2: This is a well written manuscript.

1. In the study setting line 106, could be good to mention the name of the two NGOs that you hard partner with unless they said they should not be mentioned.

2. Raw de-identified data especially the quantitative should be deposited on the public repository e.g., demographic, pre-post scores, competency, etc. Here is an example of a public space for depositing the data: https://zenodo.org/

Reviewer #3: - Can you be more concrete about how SU and depression related stigma is related to loss of follow-up? For instance, how much more likely are patients to drop out and express SU and depression stigma as a reason for their loss of follow-up vs. people who are lost to follow-up for other reasons?

- Why were these two NGOs selected specifically?

- How did you select the role play new patient? How was their training done?

- Since you’re using number between parenthesis in the methodology, i.e. Line 149-152 “According to the Link and Phelan stigma framework, mental health and SU stigma may exist among CHWs, when (1) CHWs label their patients who have SU and/or depression symptoms as different and attach negative stereotypes to them, and (2) separate these patients (‘them’) from other patients and themselves (‘us’)”,

this type of listing could be easily confused with the referencing, despite using brackets for the references and parenthesis for your listing of the steps taken. I suggest listing your steps using another format, such as

“According to the Link and Phelan stigma framework, mental health and SU stigma may exist among CHWs, when a.) CHWs label their patients who have SU and/or depression symptoms as different and attach negative stereotypes to them, and b.) separate these patients (‘them’) from other patients and themselves (‘us’),

- Maybe you should provide a bit more information for how you integrated Link and Phelan’s stigma framework and the sIMB-CIM

- One of the assumptions to us ANOVA is that your data is normally distributed. I suggest adding a sentence clarifying which normality test you used to test for normality to ensure all statistical steps were taken.

- My apologies, as non-native English speaker, I am struggling to find sense to the word “vignette” in your paper. It is my understanding vignette is more often referred to presenting a clear description, sketch, or picture of something, so I’m thinking that when you use vignette, you mean something like “participant characteristics”. Please reconsider your phrasing of this word for something simpler or closer to what you mean, as it could be confusing for some of your readers.

- In fig.1 all training days have 17, which is your total number. Training days should have the individuals who were trained for each day, summing up to 17 instead.

- Since the training was done in English, it would be helpful to detail how you controlled or minimised language barriers during the training, and/or desirability bias when the participants were answering their surveys, specially the post-training assessment.

6. PLOS authors have the option to publish the peer review history of their article (what does this mean?). If published, this will include your full peer review and any attached files.

**Do you want your identity to be public for this peer review?** For information about this choice, including consent withdrawal, please see our Privacy Policy.

Reviewer #1: **Yes: **Archana Siddaiah

Reviewer #2: **Yes: **Moses Banda Aron

Reviewer #3: No

---

## [Decision Letter · Decision Letter 1]

3 Nov 2023

Piloting Siyakhana: A community health worker training to reduce substance use and depression stigma in South African HIV and TB care

PGPH-D-23-00892R1

Dear Ms. Regenauer,

We are pleased to inform you that your manuscript 'Piloting Siyakhana: A community health worker training to reduce substance use and depression stigma in South African HIV and TB care' has been provisionally accepted for publication in PLOS Global Public Health.

Best regards,

Rashmi Josephine Rodrigues, M.D., Ph.D.

Academic Editor

Reviewer Comments (if any, and for reference):

Reviewer's Responses to Questions

**Comments to the Author**

1. If the authors have adequately addressed your comments raised in a previous round of review and you feel that this manuscript is now acceptable for publication, you may indicate that here to bypass the “Comments to the Author” section, enter your conflict of interest statement in the “Confidential to Editor” section, and submit your "Accept" recommendation.

Reviewer #1: All comments have been addressed

Reviewer #2: All comments have been addressed

2. Does this manuscript meet PLOS Global Public Health’s publication criteria? Is the manuscript technically sound, and do the data support the conclusions? The manuscript must describe methodologically and ethically rigorous research with conclusions that are appropriately drawn based on the data presented.

Reviewer #1: Yes

Reviewer #2: Yes

3. Has the statistical analysis been performed appropriately and rigorously?

Reviewer #1: Yes

Reviewer #2: Yes

4. Have the authors made all data underlying the findings in their manuscript fully available (please refer to the Data Availability Statement at the start of the manuscript PDF file)?

Reviewer #1: Yes

Reviewer #2: Yes

5. Is the manuscript presented in an intelligible fashion and written in standard English?

Reviewer #1: Yes

Reviewer #2: Yes

6. Review Comments to the Author

Reviewer #1: None

Reviewer #2: (No Response)

7. PLOS authors have the option to publish the peer review history of their article (what does this mean?). If published, this will include your full peer review and any attached files.

**Do you want your identity to be public for this peer review?** For information about this choice, including consent withdrawal, please see our Privacy Policy.

Reviewer #1: No

Reviewer #2: **Yes: **Moses Banda Aron
